# Geniposide and Harpagoside Functionalized Cerium Oxide Nanoparticles as a Potential Neuroprotective

**DOI:** 10.3390/ijms25084262

**Published:** 2024-04-11

**Authors:** Rosa Martha Pérez Gutiérrez, Luis Miguel Rodríguez-Serrano, José Fidel Laguna-Chimal, Mónica de la Luz Corea, Silvia Patricia Paredes Carrera, Julio Téllez Gomez

**Affiliations:** 1Natural Products Research Laboratory, Higher School of Chemical Engineering and Extractive Industries, National Polytechnic Institute (IPN), Av. National Polytechnic Institute S/N, Mexico City 07708, Mexico; 2Faculty of Psychology, Universidad Anáhuac México Norte, Huixquilucan 52786, CP, Mexico; l.rodriguez@anahuac.mx (L.M.R.-S.); fidel.lagunach@anahuac.mx (J.F.L.-C.); julio_tellez_gomez@hotmail.com (J.T.G.); 3Polymer Research Laboratory, Higher School of Chemical Engineering and Extractive Industries, National Polytechnic Institute (IPN), Av. Instituto Politécnico Nacional S/N, Mexico City 07708, Mexico; mcoreat@yahoo.com.mx; 4Sustainable Nanomaterials Laboratory, Higher School of Chemical Engineering and Industries Extractives, National Polytechnic Institute (IPN), Av. National Polytechnic Institute S/N, Mexico City 07708, Mexico; silviappcar@gmail.com

**Keywords:** Alzheimer’s disease, tau protein, amyloid-β1–42, nanoparticles, cognitive performance, acetylcholinesterase, butyrylcholinesterase

## Abstract

Alzheimer’s disease is associated with protein aggregation, oxidative stress, and the role of acetylcholinesterase in the pathology of the disease. Previous investigations have demonstrated that geniposide and harpagoside protect the brain neurons, and cerium nanoparticles (CeO_2_ NPs) have potent redox and antioxidant properties. Thus, the effect of nanoparticles of Ce NPs and geniposide and harpagoside (GH/CeO_2_ NPs) on ameliorating AD pathogenesis was established on AlCl_3_-induced AD in mice and an aggregation proteins test in vitro. Findings of spectroscopy analysis have revealed that GH/CeO_2_ NPs are highly stable, nano-size, spherical in shape, amorphous nature, and a total encapsulation of GH in cerium. Treatments with CeO_2_ NPs, GH/CeO_2_ NPs, and donepezil used as positive control inhibit fibril formation and protein aggregation, protect structural modifications in the BSA-ribose system, have the ability to counteract Tau protein aggregation and amyloid-β1–42 aggregation under fibrillation condition, and are able to inhibit AChE and BuChE. While the GH/CeO_2_ NPs, treatment in AD induced by AlCl_3_ inhibited amyloid-β1–42, substantially enhanced the memory, the cognition coordination of movement in part AD pathogenesis may be alleviated through reducing amyloidogenic pathway and AChE and BuChE activities. The findings of this work provide important comprehension of the chemoprotective activities of iridoids combined with nanoparticles. This could be useful in the development of new therapeutic methods for the treatment of neurodegenerative diseases.

## 1. Introduction 

The unfavorable consequence of protein aggregation on safety and health in humans can be multiple; however, there is evidence to be related to the aging process, amyloid diseases, neurodegenerative diseases, and enhanced cytotoxicity [1]. The aggregation proteins in the form of neurofibrillary tangles contribute to neurotoxicity in Alzheimer’s disease (AD), which are composed of extracellular senile plaques and hyperphosphorylated tau 1,2 deposited in small peptides of amyloid-β (Aβ) derived from sequential proteolytic cleavages of the amyloid precursor protein (APP) 3. These mutations change APP metabolism, facilitating the generation of aggregation-prone Aβ species. In this context, the “amyloid cascade hypothesis” forms the basis for AD pathogenesis [2].

Numerous studies have suggested that the learning and memory deficits related to aging and AD are attributed to the deterioration of the cholinergic magnocellular neurons of the nucleus basalis of Meynert (nbM). Degeneration of neural pathways implies the transmitters as well as serotonin, acetylcholine, dopamine, and noradrenaline [3]. In addition, a significant reduction of cholinergic markers and choline acetyltransferase has been identified in demented patients. According to this hypothesis, changes in memory and learning deficits that occur in AD and aging could be attributed to an impairment of basal forebrain cholinergic systems [4].

Alzheimer’s (AD) is a neurodegenerative disease associated with progressive cognitive decrement. Many biomarkers, such as low-density lipoprotein receptor-related protein (LRP1), amyloid precursor protein (APP), amyloid-beta peptide 1–42 (Aβ 1–42), and tau-5, are affected by epigenetic risk factors and are highly associated with AD progression [5]. In consequence, treatment of AD with multiple targets can produce a better effect than those using single-target [6].

Cholinesterase (ChEs) is responsible for the hydrolysis and metabolism of AChE to control its levels. Acetylcholinesterase (AChE) and butyrylcholinesterase (BuChE) are two important enzymes. AChE is the main enzyme that hydrolyzes Ach and has a neuroprotective effect on the stimulation of muscarinic receptors and nicotinic receptors, thus improving cognitive function. Butyrylcholinesterase (BuChE) plays a crucial role as a compensating enzyme in ongoing AD. It has been shown that B ChE is highly correlated with abnormal β-amyloid (Aβ) deposition. BuChE participates in the neural function and is distributed in the nervous system. In addition, it plays an important role in cholinergic neurotransmission and neurodegenerative diseases [7].

AD is considered an amyloidogenic disorder characterized by the accumulation and aggregation of misfolded Ab peptides. During aggregation, polymorphic species are generated from Ab monomers, yielding a heterogeneous population of transient on-pathway Ab oligomers that are eventually transformed into larger protofibrils and fibrils. The structured Ab oligomeric intermediates are suggested to be the primary toxic agents, while fibrils are indicated to be inert and relatively less toxic, but they can still act as a reservoir of toxic oligomers through fragmentation and secondary nucleation [8]. In consequence, amyloid-b (Ab) aggregation and accumulation cause neuronal death, leading to AD [9].

Glycation reaction is a process between free amino groups of the proteins and free reducing sugars. This Maillard reaction is known as advanced glycation end-products (AGEs), which are irreversible adducts that accumulate in the brain with aging. In AD individuals, the presence of Aβ deposition, microglial cells, and the increment of AGE levels possess a pathological role in this neurodegeneration disease [10].

Arduous research efforts persist in developing effective disease-modifying treatments (DMTs) for AD, as well as symptomatic therapeutics. A plethora of continuing phase 1, 2, and 3 human studies are focused on various treatment targets in AD. Consequently, anti-amyloid DMTs have focused on four major MOAs: (1) reduction of Aβ42 production (γ-secretase inhibitors, β-secretase inhibitors, α-secretase potentiation), (2) reduction of Aβ-plaque burden (aggregation inhibitors, drugs interfering with metals), (3) promotion of Aβ clearance (active or passive immunotherapy), and (4) inhibition AChE and BChE increase the availability of acetylcholine at synapses and have been proven clinically useful in delaying the cognitive decline in AD including donepezil, galantamine, rivastigmine and low-to-moderate affinity, noncompetitive N-methyl-d-aspartate (NMDA) receptor antagonist memantine [11]. Memantine can be administered in combination with an AChE; their combination benefits patients without any increase in adverse effects. Some of the recent failures of anti-amyloid agents in phase 3 clinical trials in patients with early-stage, mild, or mild-to-moderate stage AD were hemiacetal, bapineuzumab, solanezumab and in similar trials of β-secretase inhibitors (BACE) lanabecestat, verubecestat, and atabecestat, which showed a significant and dose-dependent result of reducing CSF Aβ42, but without cognitive or functional benefit, while many of them were poorly tolerated and some of them failed in subjects with prodromal AD. These results might support the suggestion that blocking the process of forming Aβ may not be capable of halting the disease progression [12]. Deferiprone is an iron-chelating agent studied in phase 2 trials in participants with mild and prodromal AD. It demonstrated promising efficacy data in preclinical studies. In a 3-month phase 2 study, PBT2 succeeded in a 13% reduction of CSF Aβ [13]. Active Aβ immunotherapy agents, such as CAD106, APOE4 homozygotes, APOE4 heterozygotes, umibecestat, anti-Aβ40 antibodies, GV1001 peptide (tertomotide), and UB-311 (synthetic peptide) [14], have been used in the clinical trial stage for AD, most of which have shown low efficacy or side effects. At the moment, used drugs for therapy of AD are only symptomatic and do not stop neuron loss.

Possibly, medicinal plants could be a better strategy for the progress of potential neuroprotective drugs for restoring normal brain functioning and ameliorating neuronal cell loss in AD patients. The therapeutic potential of medicinal plants is associated with the content of their bioactive chemical. In addition, traditional medicinal plants are widely used for the treatment of dementia-related disorders as well as memory enhancers. In previous works, it has been found that some plants possess acetylcholine esterase inhibitory activity, are cognitive enhancers, and have strong antioxidants such as *Ginkgo biloba* [15]. Many medicinal plants and their constituents are used in traditional medicine to alleviate cognitive function, depression, memory loss, and other symptoms of AD. A mixture of plants is more prescribed than a simple herb because of the bioactive compound content that could act as modulators of other constituents from the plants or could act synergistically. This concept has been used in Native Americans’ traditional and Chinese medicine systems and in Ayurveda, where a mixture of two or more plants is recommended for whatever disease is determined.

Monoterpene iridoids are widely distributed as glycosides in several families, such as Verbenaceae, Gentianaceae, Scrophulariaceae, Bignoniaceae, Rubiaceae, Loganiaceae, and Labiatae. Some iridoids include harpagoside, geniposide, coumaroylcatalpol, and catalpol, as well as several secoiridoid aglycones such as oleocanthal and oleuropein aglycone. They have shown a potent neuroprotective effect against dementia-related disorders. Acylated and aglycones iridoids crossed the BBB through passive diffusion, those in the glycated counterpart, increasing bioavailability in the brain [16]. A new strategy for the encapsulation of iridoids to improve their bioavailability shows promising results.

Geniposide is a constituent of *Gardenia jasminoides* fruits, which was administrated for 8 days to MPTP-mice; it increased the number of TH-positive dopaminergic neurons, and it increased the levels of caspase 3 proteins and pro-apoptotic signaling Bax and the SNpc mice brain, protecting its neurons and enhancing the exploratory and locomotor activities [17]. Harp is a constituent of *Scrophularia ningpoensis* and *Harpagophytum procumbens*, which significantly enhance memory function, locomotor activity, and the impaired spatial learning enhancement at the cortex hippocampus regions, the BDNF levels, and the activation of PI3K/Akt and ERK for up-regulation of BDNF levels. Harpagoside is effective as a neuroprotective agent, ameliorating memory and learning impairment, and it protects the cerebral neurons from apoptosis, fear memory, and spatial learning/memory impairments. These effects are related to the binding of BDNF protein to its receptor TrKB, the up-regulation of BDNF levels, and the activation of MAPK/PI3K kinases [18].

Nanoparticles improve bioactivity, reduce toxicity, enhance targeting, and provide a versatile manner of handling the release of the encapsulated compound. Cerium oxide nanoparticles have recently been considered for biomedical applications because of their potent antioxidant redox properties. Considering that the brain is subject to elevated levels of oxidative stress, especially in neurodegenerative disease. Previous studies have reported that cerium nanoparticle treatment could reduce the memory deficits present in rodent models and AD patients in clinic studies [19]. However, intravenous administration is not recommended owing to their deficient physicochemical properties, deficient blood–brain penetration, and rapid blood clearance [20]. There are contradictory reports on whether CeO_2_ NPs may act as an oxidant causing toxicity or as an antioxidant, being able to scavenge free radicals and protect the cells from oxidative damage. CeO_2_ NPs displayed low toxicity without increasing the number of necrotic cells compared to the control group, which can significantly inhibit the number of dead hippocampal neurons [21]. In tau-induced Drosophila models of Alzheimer’s disease, CeO_2_ NPs were reducing the levels of tau at the transcriptional level. However, the climbing effect of elevated higher concentrations produces detrimental effects [22]. Thus, another study demonstrated that 20-nm CeO_2_ nanoparticles reduce cell viability in human bronchoalveolar carcinoma-derived cells at 3.5 to 23.3 µg/mL produce significant oxidative stress in the cells, and increase the production of LDH and MDA [23]. Pulido-Reyes et al. [24] demonstrated that neither shape, concentration, surface charge (ζ-potential), synthesis method, nor nominal size had any influence on the observed toxic effects. These depend on the percentage of surface content of Ce3+ sites. CeO_2_ NPs have been shown to protect cells from reactive oxygen species due to their inherent antioxidant properties [25]; this protective effect was thought to be due to the presence of a dual oxidation state of CeO_2_ NPs. With these contrasting results, the toxicity of CeO_2_ NPs remains elusive, and specific toxicity endpoints relevant to human health need to be addressed. However, various biomedical applications of CeO_2_ NPs, such as protection against radiation-induced damage, retinal neurodegeneration, and anti-inflammatory and antioxidant activity, have also been studied [26].

The biological activity of compounds from plants depends on their functional groups, which show diverse forms of non-covalent interactions. Some limitations of these compounds are related to their poor bioavailability and low solubility. These deficiencies can be surpassed by the use of nanoparticle delivery systems because they have shown an increase in bioavailability, stability, and water solubility [27]. This research work is focused on the synthesis and neuroprotective activity of an association between CeO_2_ NPs and a combination of geniposide and harpagoside (GH/CeO_2_ NPs), and their mechanism of action for causes on vitro assays against aggregation of proteins, cholinergic deficits, and cognitive dysfunction induced-AlCl_3_ in mice model of neurodegenerative disease.

## 2. Results

### 2.1. Characterization of Encapsulation of GH into Cerium Nanoparticles

UV-visible and FTIR spectroscopy revealed the first evidence that shows the formation of cerium and encapsulation of GH, Figure 1A,B and Figure 2A,C. At the beginning of the reaction, the color of the mixture was pale yellow, which gradually was modified to intense yellow (Figure 1A). This color was associated with the excitation of the surface of the CeO_2_ nanoparticles, indicating that Ce(NO_3_)_3_ is reduced to the CeO_2_ nanoparticle. The UV-visible spectrum of native GH showed a characteristic absorption at 257 nm attributed to one double bond conjugated with an iridoidic lactone system.

Pure cerium nanoparticles exhibit a strong absorption pick in the UV region near 301 nm due to the charge-transfer transitions deriving from O 2p to Ce 4f, which exceed the f–f spin-orbital splitting of the Ce 4f state [28]. This absorption was compared with obtained peaks for Ce(NO_3_)_3._6H_2_O at 252.5 m 267 and 390 nm, which confirms the formation of CeO_2_-NPs. The UV spectra demonstrated that GH had been successfully encapsulated in the synthesized nanoparticles. The FTIR spectra of GH, CeO_2_ NPs, and GH/CeO_2_ are displayed in Figure 2A,C. The GH spectrum shows distinctive peaks at 3227 cm^−1^ (hydroxyl) and carbonyl at 1638 cm^−1^, 1462, and 1352 cm^−1^, which are characteristics of iridoids. The absence of these peaks in the GH/CeO_2_ spectra are indirect evidence of encapsulation. The interactions are generally physical in nature, and in consequence, the encapsulation does not chemically modify the groups. In addition, the presence of the peaks in the region between 1620 and 981 cm^−1^ demonstrates a total amalgamation of GH and cerium, revealing the total encapsulation of GH in cerium.
Figure 2FT-IR spectra of (**A**) GH, (**B**) CeO_2_ NPs, and (**C**) GH/CeO_2_ NPs.
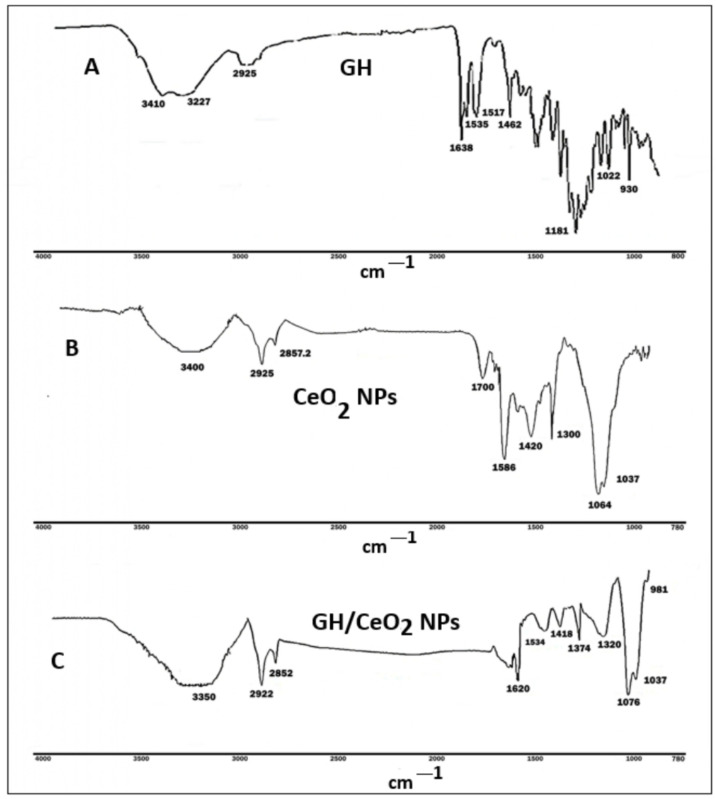


#### 2.1.1. Particle Size Distribution

Yellow elemental cerium was formed by reducing bulk cerium to its nanoform. The mean particle size, polydispersity index (PDI), and zeta potential of the cerium nanoparticles and GH/CeO_2_-NPs were measured using dynamic light scattering (DLS). The results revealed the formation of cerium nanoparticles with a size between 70.2 and 156.5 nm with an average diameter of 134.6 nm (Figure 3). The zeta potential values measured for this system were −22.3 mV, while the polydispersity index was found to be 0.240. The DLS analysis of the colloidal solution of GH/CeO_2_ NPs is shown in Figure 3B. The results indicated that biosynthesized particles have an average particle size of 94.51 nm and a zeta potential of −22.3 mV. The polydispersity index is used to describe the degree of non-uniformity of a size distribution. Different size distribution algorithms work with data that fall between these two extreme values of PDI (i.e., 0.05–0.7). Thus, values smaller than 0.05 are mainly seen with highly monodisperse standards, and values bigger than 0.7 indicate that the sample has a very broad particle size distribution DLS technique (A). GH/CeO_2_ NPs showed a PDI of 0.240 and values smaller than 0.05, confirming the degree of dispersion of the nanoparticles [29]. Considering that the zeta potential is the value of the effective electric charge on the surface of NPs provides information on particle stability. The increasing magnitude of the zeta potential augments the stability of nanoparticles due to considerable electrostatic repulsion between nanoparticles.

#### 2.1.2. Transmission Electron Microscopy, Scanning Electron Microscopy

The high-resolution TEM images of CeO_2_ nanoparticles indicate that they have a spherical shape and are moderately agglomerated (Figure 4A,B). The morphology of GH/CeO_2_-NPs showed different morphology shapes (Figure 4C,D). TEM images revealed a significant difference in chemical composition between the cerium nanoparticles untreated and treated with iridoids.

#### 2.1.3. X-ray Diffraction (XRD) Analysis of GH/Cerium Nanoparticles

The analysis of X-ray diffraction of the structure of GH/cerium nanoparticles and their characteristic peaks based on their elemental composition are observed in Figure 4C. Different elements’ peaks, including C and O_2_ and cerium, were detected. The elemental analysis indicates a peak attributed to cerium around 21.12% of the mass, confirming the transition from bulk form to nano form. The pure n phase of GH/CeO_2_-NPs is supported by the sharpness of the peaks. The largest elements identified were carbon and oxygen, with 60.09% and 18.79% (*w*/*w*), respectively. Also, XRD assays were performed to verify the bioactive (GH) encapsulation in the synthesized NPS. The results indicated that no characteristic crystalline peaks of cerium nanoparticles were shown [30]; then, encapsulation of CH in cerium NPs revealed that the bioactive was in a molecularly dispersed or amorphous state, facilitating the diffusion of iridoid molecules throughout the cerium matrix, resulting in a constant release of the bioactive from NPs.

#### 2.1.4. Raman Spectroscopy Imaging

Raman spectra of CeO_2_ NPs and GH/CeO_2_-NPs are shown in Figure 4D. Raman spectra of CeO_2_ NPs revealed a peak at 460 cm^−1^ due to oxygen atoms neighboring the cerium ions [31]. After the encapsulation with GH, the Raman spectra shifted to 449 cm^−1^ due to the tensile strain in the matrix of the cerium [31].

#### 2.1.5. Encapsulation Efficiency (EE, %) and Loading Capacity (LC, %) of GH in CeO_2_-NPs

The percentage of the GH entrapped in cerium NPs is known as encapsulation efficiency. In this work, the EE was found to be between 81.3% (1:0.5), 77.4% (1:0.75), and 69.2% (1:1), and they can be observed in Figure 5A. The highest efficiency was obtained at a cerium/GH ratio of 1:0.5. The results indicated that EE decreases as the amount of GH increases, and the excess GH cannot be adsorbed by CeO_2_ NPs; consequently, the extract will be easily released during the centrifuge. The loading capacity (LC) is the amount of the GH encapsulated per unit weight of the cerium nanoparticles and is significantly affected by the molecular weight of the compounds to be encapsulated. The loading capacity of the cerium nanoparticles ranged from between 59.0% (1:05), 37.8% (1:0.75), and 28.6 (1:1) (Figure 5B). The results demonstrated that GH/CeO_2_-NPs have an excellent delivery ability.

### 2.2. GH and GH/CeO_2_-NPs Inhibited Acetylcholinesterase (AChE) and Butyrylcholinesterase (BuChE) Activities

CeO_2_-NPs, GH, and GH/CeO_2_-NPs on AChE and BuChE activities were carried out according to the reported assay [32]. The results observed for the AChE and BuChE inhibitory activities of CeO_2_-NPs, GH, and GH/CeO_2_-NPs are shown in Table 1. Doses of 10, 20, and 30 μg each displayed dose-dependent inhibition on AChE and BuChE. The finding proves that GH/CeO2-NPs at a concentration of 30 μg exhibited moderate inhibitory activities of AChE and BuChE by 67% and 63%, respectively, compared with donepezil was used as the positive control, about 88.7% and 84%, respectively (Table 1). The inhibition of these enzymes was a little lower than expected but still the effect is acceptable.

### 2.3. Evaluation of Anti-Aggregatory Activity

#### 2.3.1. Total AGEs-Fluorescence

The fluorescent AGE formation was evaluated in a BSA-ribose system used as a glycation model and monitored throughout 2 weeks of incubation. In this experiment, a significant (*p* < 0.05) increment in fluorescent intensity was observed in BSA incubated with ribose. The finding indicated that the fluorescent AGE formation was increased 8.69-fold. This research studied the influence of GH, CeO_2_-NPs, and GH/CeO_2_-NPs on the formation of aggregated proteins in BSA on ribose incubation using Congo red, thioflavin T, thermal aggregation, and turbidity assays.

#### 2.3.2. Fluorescence Intensity Studies of BSA-Ribose with Nanoparticles

The fluorescent AGE formation was evaluated in BSA-ribose. This system was used as a glycation model and monitored throughout 4 weeks of incubation. An important (*p* < 0.05) increment in fluorescent intensity in BSA incubated with ribose was observed in this experiment. The finding indicated that the fluorescent AGE formation was increased by 8.69-fold. Ribose-modified BSA showed a higher emission fluorescence intensity of 74.8 A.U (arbitrary unit) than native BSA (8.6 A.U), while fluorescence intensity decreased with the addition of GH, CeO_2_-NPs, and GH/CeO_2_-NPs in 42.6 A.U, 50.2 A.U, and 35.1A.U., respectively (Table 2).

#### 2.3.3. Dynamic Light Scattering

Zeta potential can measure the aggregation, surface hydrophobicity, and conformational change of a macromolecule. The values of the zeta potential of native BSA were found to be −8.41 mV, while glycated BSA showed a value of −34.6 mV (Table 2). However, glycated BSA in the presence of CeO_2_-NPs and GH/CeO_2_-NPs exhibited an important reduction in zeta potential values, which were higher than donepezil. Increasing the incubation time produces light scattering that is high in the hydrodynamic sizer of the molecules in the solution, indicating that aggregation is produced. Table 2 shows that the increment of diameter is less important for nanoparticles than for highly glycated albumin. Findings indicated that nanoparticles had a stronger inhibition effect on aggregate formation from the BSA-ribose model (Table 2). Considering that the polydispersity indexes indicated the homogeneity of dispersion, values of 0.1 or lower are ideal; instead, values above 0.1 indicate that more species are present in the solution [32]. In our sample, we obtained values of PDI of less than 0.1, indicating that the nanoparticles were near mono-disperse, Z-average sizes.

#### 2.3.4. Determination of Amyloid Aggregation by Thioflavin T

Thioflavin T is a benzothiazole dye with a low molecular weight compound specific for detecting intermolecular β-sheets’ content in amyloid/aggregated structures of proteins. Thioflavin T binds with protein structures and displays a characteristic green fluorescence by the aggregated proteins, and the intensity is directly proportional to the degree of aggregation. In this study, the fluorescence intensity of thioflavin T was significantly elevated in glycated BSA, evidencing the formation of aggregated structures, while the native BSA showed little fluorescence when binding with ThT. The thioflavin T fluorescence experiment in glycated BSA treated with GH, CeO_2_-NPs, and GH/CeO_2_-NPs were performed to measure the development of aggregated structures. The reduction of the ThT fluorescence implies a protective effect on the development of aggregates; the results indicated that GH and GH/CeO_2_-NPs avoid the formation of the aggregates. In addition, GH/CeO_2_-NPs were revealed to display more anti-aggregation effects than GH and donepezil (Figure 5C). The findings demonstrated that the treatment with GH, CeO_2_-NPs, and GH/CeO_2_-NPs reduced by 69, 65, and 89%, respectively. Donepezil showed lower inhibition than nanoparticles (47%). Consequently, NPs have the ability to avoid the development of aggregates in glycated BSA. In addition, GH/Se-NPs were revealed to show more anti-aggregating effects than other treatments.

#### 2.3.5. Determination of Amyloid-Beta Aggregation by Congo Red Assay (CR)

The Congo red assay was carried out to support the prime feature of protein aggregates (formation of β-sheet structures). The technique was performed to complement the results found in the ThT fluorescence assay regarding the production of BSA aggregates on sugar incubation. Congo red displayed an absorbance of around 490 nm, while on binding with the β-sheet structure of amyloid fibrils, absorbance shows a shift from 490 to 540 nm [29]. The Congo red assay of glycated samples in the presence of 3 mg resulted in the inhibition of β-amyloid aggregates in the CeO_2_-NPs, GH, and GH/CeO_2_-NPs to 81.3%, 70.2%, and 91.2%, respectively, when it was compared to glycated BSA at 30 days. The glycated BSA with GH/CeO_2_-NPs (2 mg) displayed a similar spectrum position to that observed in native BSA (Figure 5D). These results agree with the data obtained in the thioflavin T-test (Figure 5C).

#### 2.3.6. Measure of Aggregation of BSA by Turbidity and Thermal Assays

The aggregation of BSA was also validated with UV absorption spectra of Congo red and thioflavin T assays on the generation of BSA aggregates; turbidity assay was performed at 450 nm. The degree of aggregation of BSA solutions in the absence and presence of glucose was monitored for turbidity measurements. An increment of turbidity was observed after day 22, achieving a maximum around day 30, suggesting the formation of aggregates (10.8-fold). This study observed that native protein had a negligible change in absorbance, demonstrating the absence of aggregates, implying that BSA is found in natural conformation. The effect of CeO_2_-NPs, GH, and GH/CeO_2_-NPs on the absorbance of glucose-incubated BSA at 450 nm are observed in Figure 5E. After day 30, the turbidity of the sample for GH, CeO_2_-NPs, and GH/CeO_2_-NPs showed an anti-aggregative effect. This effect on turbidity was significant for GH/CeO_2_ NPs treatment, which was followed by other treatments. Findings indicate that nanoparticles prevent aggregate formation and hinder their formation (Figure 5E).

Similarly, the inhibition of β-amyloid aggregates in the thermal assay was observed at 3 and 6 h incubation (Figure 5F). At 6 h of incubation, the β-amyloid aggregates were reduced to 93% by GH/CeO_2_-NPs, 75% by CeO_2_-NPs, and 86% by GH. Thus, treated glucose BSA displayed more aggregation than native BSA, indicating that nanoparticles inhibited the B-amyloid aggregate formation in glycation development and thermal aggregation.

#### 2.3.7. Amyloid-β1–42 Aggregation

This experiment produced Aβ42 aggregates in vitro, and the ability of GH, CeO_2_-NPs, and GH/CeO_2_-NPs were determined to neutralize amyloid aggregation, employing ThT assay as an indicator in vitro of the content of amyloid fibrils [33]. The finding demonstrated the formation of Aβ42 aggregates after 12 h incubation, indicated by the increase in fluorescence intensity (Table 3). However, GH, CeO_2_-NPs, and GH/CeO_2_-NPs interfere and interact with the formed structures during the process of aggregation (Table 3). GH/CeO_2_-NPs inhibit 91% of the aggregation process at 24 h of incubation, compared with GH and CeO_2_-NPs with 84% and 81% of inhibition in the aggregation of amyloid-β1–42, respectively. Consequently, in vitro assay GH/CeO_2_ could reduce the fibrillar Aβ formed, supporting that nanoparticles have the ability to protect against Aβ toxicity.

The finding suggested that the capacity of iridoids to decrease amyloid aggregation could be due to the presence of hydroxyl groups forming hydrogen bonds with β1–42.

#### 2.3.8. Tau Protein Aggregation

Aggregation of tau protein is usually performed through the nucleation based on fibril polymerization in three stages: (i) activation and nucleation (lag phase), (ii) elongation (growth phase), and (iii) steady phase (PHFs) [33]. Thioflavin T assay employing heparin as an inducer was used to measure the effect of GH, CeO_2_-NPs, and GH/CeO_2_-NPs on the disaggregation of Tau aggregates. Heparin, a polyanionic cofactor, induces nucleation for β-structure production; then it is accumulated to produce mature tau fibril as is shown in Table 4 when GH, CeO_2_-NPs, and GH/CeO_2_-NPs are added to tau protein and incubated for 120 h. GH, CeO_2_-NPs, and GH/CeO_2_-NPs have the ability to bind to intermediate structures of tau protein; the decrement of tau aggregates in a dose-dependent manner suggests the ability of nanoparticles to avoid the conversion during the fibrillation process to more aggregated conformations, which is more prominent for GH/CeO_2_ NPs treatment those in the GH and CeO_2_ NPs.

### 2.4. Effect of CeO_2_-NPs, GH, and GH/CeO_2_-NPs on Behavioral Condition in Treated AlCl_3_ Mice 

The goal was to investigate if CeO_2_ NPs, GH, and GH/CeO_2_ NPs can recover memory loss and cognitive performance. For that, a treatment with an AD-induced-AlCl_3_ mouse with CeO_2_-NPs, GH, and GH/CeO_2_-NPs daily for 7 weeks was carried out to determine spatial learning and memory deficits. The exploratory and locomotor behavior was measured in an open field and object recognition tests (Figure 6A–C). T-maze was used to examine the mice’s open field and object recognition tests (Figure 6A–C). T-maze was used to examine the mice’s memory and cognitive abilities in space exploration and evaluate the protective effect of CeO_2_-NPs, GH, and GH/CeO_2_-NPs on common behavioral dysfunction in AlCl_3_-induced memory impairment mice. The AlCl_3_-induced AD (control group) showed a 29% space exploration rate for the new routes, while the space exploration of the new route was 43%, 50%, and 70% for CeO_2_-NPs, GH, and GH/CeO_2_-NPs groups, respectively. Instead, the group was treated with donepezil, which is an acetylcholinesterase inhibitor used as a positive control, displayed a new route exploration rate of 59%. However, during the training period of the T-maze test on AlCl_3_-mice treated with CeO_2_-NPs, GH, and GH/CeO_2_-NPs, they were shown to enhance memory retention during the time to arrival. The comparison of the covert platform with AlCl_3_-mice and the normal group (Figure 6A) demonstrated that any group that was impaired in exploratory and motility activities improved the cognitive function and decreased the time taken to complete the task in the AD mice.

#### 2.4.1. Nanoparticles Enhanced Cognitive Abilities of AD Mice in an Open-Field Test

In this research, mice with AD induced by AlCl_3_ treatment were employed to study the effects of CeO_2_-NPs, GH, and GH/CeO_2_-NPs. The AD mice in the open-field test showed a major number of disordered movements, without any purpose, around the central field region of those in the control group (Figure 6B). In C57BL6/J mice with AD, nanoparticles significantly ameliorated locomotor activity compared with AlCl_3_-administered mice. The improvement of the animals’ exploratory movement, memory, and cognition abilities is also well supported by results obtained in the T-maze test.

#### 2.4.2. Effect of CeO_2_-NPs, GH, and GH/CeO_2_-NPs on Object Recognition Ability

Throughout the training session, the rodent did not show significance for the recognition of two objects (A, A′). Then, one of the objects was changed, after 24 h, toward two different objects (A, B). In the normal and AD control group, the curiosity of familiar and novel objects was shown in Figure 6C. The normal control group showed less exploration of the familiar object compared with the novel object, while the AD group indicated no statistically significant difference between the familiar and novel objects, demonstrating cognitive impairment in the AlCl_3_ mouse. However, the animals treated with CeO_2_ NPs, GH, and GH/CeO_2_ NPs displayed an improvement in the cognitive ability toward novel objects. In particular, the GH/CeO_2_ NPs group showed an increase in curiosity toward novel objects compared to those in the other groups.Figure 6Effect of GH, CeO_2_ NPs, and GH/CeO_2_ NPs to determine behavioral status in experimental AD mice. (**A**) T-maze tests show spatial perceptive ability as a percentage of the number of arm visits for 5 min; (**B**) total squares crossed (5 min) in the open field. In the novel object recognition test, the space perceptive ability is determined as the percentage of the identification of a rodent in each object for 5 min (**C**). In the object recognition test, abilities for familiar and new objects are significantly different as determined by the Student’s *t*-test (* *p* < 0.05). (**D**) ELISA test reducing Aβ deposition after I GH/CeO_2_ NPs treatment in plasma of AlCl3 mice. Effect of samples on plasma amyloids Aβ1–42 (**D**). Data are expressed as the mean ± SD *** *p* < 0.001, ** *p* < 0.01, and * *p* < 0.05 compared with the AlCl_3_ group.
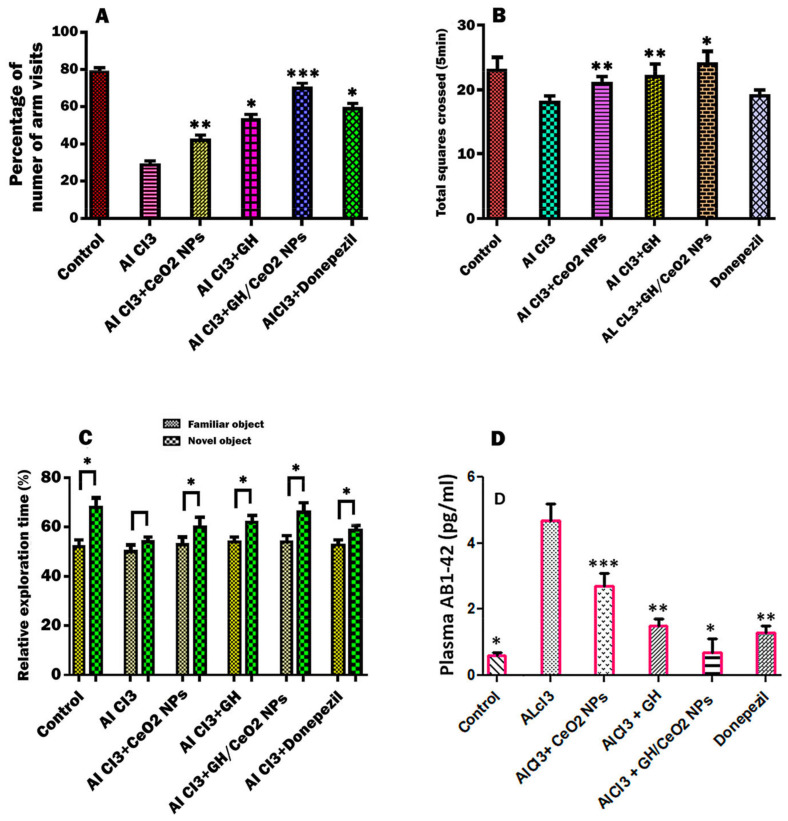


#### 2.4.3. Nanoparticles Treatment Reduces Plasma Aβ Deposition in C57BL6/J Mice

Considering that Aβ deposition is a main event in AD disease. For this reason, an ELISA test was carried out to determine Aβ1–42 levels in the plasma. The levels of amyloids in untreated mice were higher compared to the control group. The treatment with nanoparticles for 7 weeks on the level of Aβ1–42 in AlCl_3_-mice with GH/CeO_2_-NPs remarkably reduced insoluble Aβ1–42 deposition similarly to the normal group (Figure 6D), while the donepezil treatment moderately reduced (28.6%) the level of Aβ1–42 in AlCl_3_-mice. The findings demonstrated that the deposition of Aβ1–42 in *C57BL6/J* mice could be an improvement in GH/CeO_2_ NP treatment.

#### 2.4.4. Cytotoxicity of CeO_2_ NPs and GH/CeO_2_ NPs

Treatment of RAW 264.7 cells with CeO_2_ NPs and GH/ CeO_2_ NPs for 24 h did not show any significant toxicity at any of the doses studied with 0, 18.7, 37.5, 75, 150, 300, and 400 µg/mL (Appendix A).

## 3. Discussion

The goal of this work was to investigate CeO_2_-NPs, GH, and GH/CeO_2_-NPs for promising levels of neuroprotection, and its hypothesized mechanism of action is focused on in vitro assays and mouse models of neurodegenerative disease. CeO_2_-NPs and GH/CeO_2_-NPs are characterized using different spectroscopic techniques. This study displayed the synthesis of GH/CeO_2_-NPs, which exhibited a nano-size, stability, amorphous nature, and spherical shape. The hydrodynamic size and stability of GH/CeO_2_ NPs were evaluated by DLS and zeta potential analysis. Zeta potential is the value of the effective electric charge on the surface of NPs and provides information on particle stability. Increasing the magnitude of zeta potential increases the stability of nanoparticles by electrostatic repulsion between nanoparticles. UV-visible, FTIR, Raman spectroscopy, and XRD assays indicated that GH was encapsulated into CeO_2_-NPs. This research provides new insight into using nanoparticles in medicine containing cerium and phytocompounds (geniposide and harpagoside) against protein aggregation for in vitro tests and AD models. Based on previous studies, CeO_2_-NPs have potent antioxidant and redox properties [34], and the obtained results indicated that they could be promising for the therapy of diseases where oxidative stress plays a critical role, such as neurological disorders. Thus, CeO_2_-NPs, by their powerful free radical scavenging ability, were employed as carriers to load geniposide and harpagoside for the treatment of AD. The compounds of natural sources have not been a favorable development as drugs due to poor bioavailability. The iridoid glycosides had the best profile in the in vitro test, indicating that they can be recommended as drug candidates. Numerous pharmacological investigations have found that iridoids have important therapeutic activities on neurodegenerative diseases through a multifactorial mechanism, including an antioxidant effect [35].

Free radicals (ROSs) contribute to the progress of neurodegenerative diseases such as AD by reducing mitochondrial function, decreasing the antioxidant system, and modifying metal homeostasis. In consequence, they alter neurotransmission in neurons and synaptic activity, leading to cognitive dysfunction [36].

AChE and BuChE participate in the hydrolysis of the neurotransmitter acetylcholine (ACh). AChE is extremely selective for acetylcholine hydrolysis, and BuChE metabolizes other different substrates and is considered less active than AChE for the regulation of Ach [37]. A large number of studies have reported that AD disease significantly increases the levels of AChE and BuChE in the brain, generating neuronal degeneration at different stages of AD, and both cholinesterases can accelerate Aβ aggregation. Restoration of the cholinergic function of AChE and BuChE is still considered an option to ameliorate cognitive performance due to the participation of different stages in the modulation of the neurotransmitter, which are considered important targets in patients with cholinergic deficits, including AD and other neurological disorders. AD patients present a link between the deposition of fibrillar β-amyloid (Aβ) in the cerebral cortex; consequently, this enzyme has an important role in the symptomatic therapy of AD [38]. Donepezil is a representative AChE inhibitor that improves damaged nerve endings in communicating with other nerve cells, which is why it is used in this study as a positive control. Also, the effect of GH/CeO_2_-NPs in Ellman’s colorimetric assay was studied to evaluate the expression of AChE and BuChE enzymes. Findings indicate that GH/CeO_2_-NPs significantly inhibit both enzymes. In addition, the observed results were less for donepezil at the same concentration, suggesting that the mechanism (s) of action implicated in the activity of the nanoparticles could be in part similar to the mechanism of donepezil, and another type of mechanism could contribute to ameliorate the disease. The results indicated a better inhibitory activity with donepezil compared to other samples.

Glycation can cause protein crosslinking and aggregation, leading to β-amyloid fibrillation. Accumulation of amyloid-beta fibrillation adducts promoted the pathological progression in neurodegenerative disorders. Based on this evidence, we have used the combination of geniposide and harpagoside with cerium nanoparticles for promising levels of neuroprotection for the first time. Their hypothesized mechanism of action was focused on in vitro and in vivo assays. A series of techniques were carried out and tested to evaluate the structural changes related to in vitro aggregation in incubation with ribose. The aggregation protein accumulates in tissues mainly by thermal denaturation, metabolic reaction, and interaction with drugs, which are found in neurodegenerative and metabolic disorders. This work investigated the effect of nanoparticles in ribose-induced aggregation on BSA structure. The fluorescence intensities of exhibited glycated BSA were about 8.98-fold that of the native BSA, which results in amyloid aggregation, which is closely associated with AD. The addition of ribose produces an increment in the β-sheet conformation in BSA. In addition, high plasma amyloid-beta concentrations are related to an increase in the development of AD disease. Usually, modified proteins by glycoxidation reaction produce alterations in their electrophoretic mobility, isoelectric point, and surface hydrophobicity. However, glycation partially unfolds proteins, modifying the electrophoretic properties. In this work, GH/CeO_2_-NPs prevent BSA glycation due to the ability of extract and CeO_2_-NPs to protect the structural modifications and surface hydrophobicity in protein as well as scavenge free radicals properties. The findings are consistent with those of the zeta potential, polydispersity index, and hydrodynamic sizer.

However, the inhibition of fluorescence intensity in incubated samples with nanoparticles compared to native BSA clearly demonstrated the inhibition of aggregate formation. The glycation of BSA in the presence of sugar, determined by thermal and turbidity assays, resulted in the aggregation of the amyloid-beta structure, which was reversed with GH/CeO_2_ NP treatment, suggesting the inhibition of larger aggregates formed. These observed findings indicate that nanoparticles effectively inhibit glycation-induced aggregation.

In the case of Aβ1–42, the presence of nanoparticles in experimental incubation conditions resulted in a significant decrease in fluorescence intensity, while Aβ1–42 untreated displayed high fluorescence intensities, indicating an important conformational modification of the peptide. Results suggested that the decrease in fluorescence intensities could be due to the quenching of amino acid residues by nanoparticles. This interaction produces a reducing effect of nanoparticles on Aβ1–42 than donepezil and, possibly, could discontinue in β-sheets the fibril-stabilizing bonds. The change in the amino acids inside amyloidogenic places plays an important role in the reduction of amyloid formation [39].

In this work, it was determined that GH/CeO_2_-NPs reduce tau aggregation, which is consistent with the decrease of amyloid Aβ1–42 accumulation. The therapeutic effects of GH/Se-NPs on amyloid aggregation can possibly interact with tau protein through hydrophobic forces, hydrogen bonds, and van der Waals forces inhibiting the lag phase of heparin-induced-tau protein aggregation [40]. In addition, the chemical structure of iridoids with their carbonyl groups could combine with lysine residues and damage the self-assembly process of nucleation and elongation of fibril formation, avoiding protein aggregation.

The aggregation of proteins, tau oligomers and amyloid-β oligomers, acetylcholine, and butyrylcholinesterase are considered the main cause of mitochondrial dysfunction, producing neuronal toxicity resulting in Alzheimer’s disease. Our work observed that the induction of a β-amyloid structure in the BSA-ribose system was suppressed in the presence of CeO_2_-NPs, GH, and GH/CeO_2_-NPs. In consequence, tau oligomers and amyloid-β oligomers, by improving biocompatibility and biodegradability, increasing the therapeutic efficacy.

The accumulation of AlCl_3_ in the brain is related to oxidative stress and causes oxidative damage to membrane lipids [41]. In part, the effect of the neuroprotector of the nanoparticle treatment was associated with strong free radical scavenging properties [15]. In this research, we studied the behavioral modification produced by chronic aluminum exposure and the effect of nanoparticle treatment employing three behavioral assays: T-maze, open field, and object recognition ability tests. ALCl_3_-induction of AD mice has been described as producing spatial memory deficits that can be evaluated in the T-maze test. In this investigation, the ALCl_3_ group was used as a control and showed a markedly difference compared with the normal control group. Thus, in the spatial exploration of the new route, the GH/CeO_2_ NPs group produced a high spatial exploration rate for the new routes in the ALCl_3_ and control groups. In addition, the GH/CeO_2_ NPs group indicated a higher exploration rate than the donepezil-treatment group. At the concentrations used in this work, no toxic effects were observed. Thus, findings indicate that nasal treatment of GH/CeO_2_ NPs has a protective activity on spatial memory deficits in C57BL6/J mice induced with AD. The novel object recognition test in an impairment AlCl_3_-induced mice model confirmed the neuroprotective effect of nanoparticles. The object recognition ability test is a task founded on spontaneous behaviors used in impaired rodents to evaluate novel object recognition. Animals are then exposed to familiar objects, setting a novel object to spend less time exploring familiar objects than novel objects [42]. In our investigation, nasal administration of GH/CeO_2_ NPs to AD mice causes a higher perception of novel objects than the control group, supporting a markedly neuroprotective effect improved in cognitive impairment. The efficiency of the animal in the behavioral tasks, specifically on motor function, was determined in an open-field test conducted to evaluate if nanoparticle treatment altered the motor function of the mice. Findings demonstrated that chronic administration of nanoparticles for 7 weeks significantly improved motor function compared to untreated AD mice, causing a significant amelioration in the acquisition and retention of knowledge for the recommended task, as evidenced by an enhancement in the behaviors, demonstrating a nootropic function. In the present research, we administered GH/CeO_2_ NPs for ALCl_3_ mice to improve AD pathology. Ameliorating memory and spatial learning of AD mice, contributing to the reduced plasma Aβ deposition.

## 4. Materials and Methods

### 4.1. Chemicals and Materials

Sigma Aldrich (St. Louis, MO, USA) purchased the metal precursor cerous nitrate, as well as the other chemicals required for the preparation of nanoparticles.

#### 4.1.1. Synthesis of CeO_2_-NPs

The CO_2_-NPs were synthesized by the sol-gel method. Briefly, 6.3 g of cerium nitrate hexahydrate (Ce (NO_3_)_3_·6H_2_O (Sigma Aldrich, St. Louis, MO, USA) was dissolved in 100 mL of Milli-Q water, and the mixture was stirred for 30 min. Then, 10% NH_4_OH was added drop by drop, maintaining a pH between 7 and 8. Then, it was vigorously stirred continuously at room temperature for 24 h. The solution was centrifuged at 20,000 rpm for 30 min to remove the excess cerium nitrate and other precursors. The supernatant was discarded, and the pellets were washed with Milli-Q water. The solution containing CeO_2_ nanoparticles was dialyzed against Milli-Q water for 24 h and stored at room temperature. Initially, the reaction mixture turned into a light-yellow color, and then the appearance of an intense yellow color confirmed the synthesis of CeO_2_-NPs.

#### 4.1.2. Encapsulation of GH into Cerium

A GH (water:ethanol 7:1) solution was added dropwise to the aqueous solution of CeO_2_ NPs. This mixture was subdued to mild stirring and kept at 55 °C for 30 min to ensure total encapsulation. The solution was ultrasonicated for 2 cycles of 300 s each, with a cooling period of 100 s for size reduction. The sonication produces the formation of homogenous nanoparticles and reduces agglomeration. The resulting mixture was washed with water:ethanol:1 *v*/*v* to eliminate the free GH until the filtrate had a pH of 7.0. The dried nanoparticles were obtained using lyophilization.

#### 4.1.3. Characterization of CeO_2_ and GG/CeO_2_ Nanoparticles

The size and polydispersity of the CeO_2_-NPs and GH-loaded cerium nanoparticles were analyzed by dynamic light scattering (DLS). The zeta potentials of CeO_2_-NPs and GHCeO_2_-NPs were measured using a Zetasizer Nano ZSP (Malvern Instruments Co. Malvern, UK). The surface morphology distribution of selenium nanoparticles were analyzed using a high-resolution transmission electron microscope (TEM; Model HT7700, Hitachi High-Tech, Tokyo, Japan). The infrared spectrum of nanoparticles was obtained by a spectrophotometer FTIR-TENSOR27 (Bruker, Ettinger, Germany) in the 400–4000 cm^−1^ range. The UV-visible spectra were obtained using a UV-1800 Shimadzu spectrophotometer (Tokyo, Japan). X-ray photoelectron spectroscopy (XPS) was performed using a spectrometer with a monochromatic Al Kα X-ray source (Thermo Fisher ESCALAB 250xi, Waltham, MA, USA). The Raman spectra of CeO_2_ NPs and GHCeO_2_-NPs were measured using a Raman analyzer from Bruker (Billerica, MA, USA).

#### 4.1.4. Encapsulation Efficiency (EE, %) and Loading Capacity (LC, %) of GH in CeO_2_ NPs

GH-loaded CeO_2_-NPs were carried out according to the earliest research [24]. Briefly, GH-loaded CeO_2_-NPs were dissolved into MeOH and stirred at 600 rpm for 3 h. Then, the mixture was centrifugated at 10,000× *g* for 1 h (1). The supernatant was collected, and the content of the unencapsulated iridoid mix (1:1; GH) was calculated from a calibration curve using a UV-1800 spectrophotometer (Shimadzu Corporation, Kyoto, Japan) at 257.5 nm. The EE% and LC% of GH were determined by using the following equations [43]:(1)$$\mathrm{Encapsulation}\;\mathrm{efficiency}\;(\mathrm{EE}\%)=\frac{\mathrm{Amount}\;\mathrm{of}\;\mathrm{encapsulated}\;\mathrm{GH}}{\mathrm{The}\;\mathrm{total}\;\mathrm{amount}\;\mathrm{of}\;\mathrm{GH}}\times 100\%$$
(2)$$\mathrm{Loading}\;\mathrm{capacity}\;(\mathrm{LC}\%)=\frac{\mathrm{Amount}\;\mathrm{total}\;\mathrm{GH}\;\mathrm{entrapped}}{\mathrm{The}\;\mathrm{total}\;\mathrm{Cerium}\;\mathrm{NPs}\;\mathrm{weight}}\times 100\%$$

#### 4.1.5. Preparation of In Vitro Glycated Albumin (BSA)

In this work, BSA was used because its structure is widely used in scientific investigations as a model protein for exogenous and endogenous ligand binding. Briefly, bovine serum albumin (BSA) (10 mg/mL) was incubated with glucose (0.5 M; 0.1 M PBS; pH 7.4) containing sodium azide (NaN_3_; 0.02%) under sterile conditions at 37 °C for 15 days. The blank test contained BSA only. The solution was subjected to dialysis with a one-time exchange of PBS in 24 h. The formation of advanced glycation and products (AGEs) were confirmed by evaluating the fluorescent in glycated albumin samples and of the positive control at excitation and emission wavelengths of 370 and 440 nm (Perkin Elmer luminescence spectrometer LS50B, Waltham, MA, USA).

#### 4.1.6. Dynamic Light Scattering Used to Determine the Aggregation of BSA

Dynamic light scattering (DLS), using a Zetasizer Nano ZSP (Malvern Instruments) by determining the growth of macromolecular assemblies through aggregation partway in the solution of BSA and BSA incubated with ribose, BSA-ribose + CeO_2_ NPs, BSA-ribose + GH, BSA-ribose + GH/CeO_2_ NPs, and donepezil at 30 days of incubation at 37 °C, determined the aggregation of proteins given for the fluorescence, size distribution, zeta potential, and polydispersity index.

#### 4.1.7. Acetylcholinesterase (AChE) and Butyryl Thiocholine (BuChE) Inhibition

Acetylcholinesterase and butyryl thiocholine inhibition were assayed using a UV-visible spectrophotometer at 412 nm. The assay buffer was NaCl (100 mM), sodium phosphate (100 mM), pH 7.3. Enzyme stock solutions were prepared with the buffer in the following concentrations: 100 U/mL and 3 U/mL, kept at 0 °C. Then, appropriate dilutions of the AChE (0.35 mM) or butyryl thiocholine iodide solutions (0.5 mM) were carried out. Stock solutions of the nanoparticles and GH were prepared in acetonitrile (10, 20, and 30 μg). After, in a cuvette containing acetonitrile (50 μL), assay buffer (830 μL), 5,5′-dithiol-bis-(2-nitrobenzoic acid) (DTNB), (50 μL), a nanoparticle solution (10 μL), and an enzyme solution (10 μL) were added and incubated for 15 min at room temperature. After mixing and bringing the absorbance to zero, the substrate (acetylthiocholine at 0.03 to 0.16 mM) was added to initiate the reaction. Assays were carried out at room temperature over a 1 min period, taking readings every 5 min.

### 4.2. Aggregation Assays

#### 4.2.1. Determination of Amyloid-β Aggregation by Thioflavin T (Th. T)

For the evaluation of the amyloid cross β-structure, another thioflavin T marker for β aggregation was used. Glycated BSA samples and positive control (100 μL) were incubated with 32 μM Th. T for 1 h. at 25 °C The fluorescence was determined at excitation and emission wavelengths of 435 and 485 nm, respectively, with correction for background signals without Th T. The results were expressed as % inhibition and were calculated by the following formula:% inhibition = (F_0_ − F_1_)/F_0_] × 100(3)
where F_0_ is the fluorescence of the positive control. F_1_ is the fluorescence of the glycated albumin samples incubated with nanoparticles.

#### 4.2.2. Binding of Congo Red

Amyloid cross β-structure, a common marker for protein aggregation, was measured using a Congo red test. For this purpose, the glycated BSA samples (50 μL) and positive control were incubated with 50 μL Congo red (100 μM in 10% (*v*/*v*) ethanol/PBS) for 20 min at room temperature. The absorbance was measured at 530 nm and recorded for the Congo red background as well as for Congo red incubated samples. The results were expressed as % inhibition calculated by the following formula:% inhibition = [(A_0_ − A_1_)/A_0_] × 100(4)
where A_0_ is the absorbance at 530 nm of positive control. A_1_ is the absorbance at 530 nm of the glycated albumin samples incubated with nanoparticles.

#### 4.2.3. Turbidimetric Aggregation Analysis

Aβ aggregation was measured by turbidimetric analysis [44] in the solution of BSA and BSA incubated with ribose, BSA-ribose + CeO_2_-NPs, BSA-ribose + GH, and BSA-ribose + GH/CeO_2_-NPs during 30 days of incubation at 37 °C. The absorbance intensity of control and sugar-incubated samples was carried out at 450 nm on a Shimadzu UV spectrophotometer UV-1800.

#### 4.2.4. Thermal Aggregation

Samples of BSA-glucose (10 mg each) were incubated alone or with nanoparticles (1, 2, and 3 mg/mL) for 3 and 6 h at 60 °C. An aliquot at 3 and 6 h was collected to analyze structural modification and aggregation. After that, fluorescence was determined at an emission of 440 nm with an excitation wavelength of 355 nm.

#### 4.2.5. β1–42 Fibrils Formation by ThT Assay

The aggregation of the amyloid was evaluated using a Th-T fluorescence assay. β1–42 was dissolved in NH_4_OH (0.1%) at 250 mM in 1,1,1,3,3,3-hexafluoro-2-propanol. This solution was diluted 10-fold with 50 mM of phosphate buffered saline (PBS; pH 7.4) and incubated at 0, 4, 6, 12, and 24 h, with or without samples (100 µg), at 37 °C. Then, 2.5 μL of the solution was added to 250 μL of Th-T (1 mM) in Gly-NaOH (50 mM; pH 8.5). The fluorescence intensity was evaluated at an excitation wavelength of 420 nm and an emission wavelength of 485 nm. Quercetin was used as a reference compound for this assay. The % inhibition of aggregation was calculated using Equation (3).

#### 4.2.6. Aggregation of Tau Proteins in Vitro

The capacity of tau protein polymerization in the presence of CeO_2_-NPs, GH, and GH/CeO_2_-NPS was evaluated at different concentrations of CeO_2_-NPs, GH, and GH/CeO_2_-NPS (0, 10, 20, and 30 µg), and they were incubated with a tau protein solution at 20 µM in 50 mM Tris-HCl (PH 7.5). Heparin (5 mM) and 1,4-ditriotreitol (DTT; 5 mM) were added to the samples, which were protected from light. Then, they were gently shaken at 180 rpm for 72 h at room temperature. Tau protein aggregation was measured by using the thioflavin T assay.

### 4.3. Behavioral Studies

#### 4.3.1. Animals of Experimentation

C57BL6/J mice (males) 16 weeks old and with 30–38 g of body weight was used in this research. Animals were maintained at 25 ± 2 °C, with a relative humidity of 45 ± 10%, a 12 h light/dark cycle, and free eating and drinking. All mice procedures were approved by the Ethics Committee of the Escuela Nacional de Ciencias Biológicas—IPN, with the principles of the Committee on Care and Use of Laboratory Animals (NIH publications 85-23, revised 1985), and comply with Mexican Official Normativity (NOM-062-Z00-1999).

#### 4.3.2. Experimental Protocol

The formed groups (n = 6) were subjected to different treatments. AlCl_3_ solution was obtained by dissolving the salt of AlCl_3_ in Milli-Q water adjusted with phosphate buffer saline (PBS; pH 7.4). This stock solution was carried out to obtain the applied dose equivalent to 100 mg/kg of AlCl_3_. This dose was chosen on the basis of previously published investigations [45]. Group (1) normal; Group II AlCl_3_ treatments were administered by oral gavage once per day for 7 weeks. Group IV AlCl_3_ + 30 mg/kg of CeO_2_ NPs; Group V AlCl_3_ + 30 mg/kg of GH; Group VI AlCl_3_ +30 mg/kg of CH/CeO_2_ NPs; Group VII AlCl_3_ + 30 mg/kg of donepezil. The samples were given daily by nasal administration, while donepezil was given daily through oral gavage. Treatment with all samples was given for seven weeks.

#### 4.3.3. T-Maze Test, Novel Objection Recognition (NOR) Test

The spatial learning and memory were evaluated by employing a T-maze test build that conformed to the protocols [46]. The T-maze assay consists of two trials carried out in an interval of 1 h. Animals were permitted to solely explore the T-maze’s start arm and familiar arm of the T-maze, while the arm containing food (third arm) was blocked. In the second test, animals were placed in the starting arm and allowed to the three arms freely for 5 min. The total time drained in the third arm was graven, carried out by a ceiling-mounted camera, and then analyzed.

#### 4.3.4. Novel Objection Recognition (NOR) Test

The NOR test consists of the spontaneous propensity of rodents to show more interactions with a novel object than a familiar thing. In the habituation phase, animals can freely explore the behavioral empty arena (50 cm × 50 cm × 25 cm) for 5 min one day before experimenting. After two objects of the same size, texture, color, and shape were placed in the arena; the animals were left to freely explore the two objects for 10 min. In experiment one, the object was replaced with a new one in the original location and repeated for 10 min; the same procedure was used. This test was conducted to evaluate the object recognition ability through sniffing and time-drained exploration of each object.

#### 4.3.5. Exploratory Behavior in Open-field Test

Exploratory behaviors were evaluated by employing an open-field build conforming to the protocols [47]. The mice were placed in the center of the open-field arena. Animals were permitted to explore the open field for 5 min. On the assay day, animals were placed in the established central position in the box, and their moving track and the time of exploring the central area within 5 min were observed and recorded. The behavior of each animal was recorded using a video camera. Olfactory odors between experiments were reduced, and the open field was cleaned with alcohol (70%). Locomotor behavior was evaluated by counting the total number of squares crossed.

#### 4.3.6. ELISA Detection of Amyloid Aggregates in Plasma

Aβ1–42 levels in plasma were tested after treatment of nanoparticles for 7 weeks, using a sensitive sandwich ELISA kit (Wako Chemicals, Richmond, VA, USA) according to the manufacturer’s instructions. The data were recorded at 450 nm using a microplate reader (Biorad, Hercules, CA, USA).

#### 4.3.7. Cytotoxicity of GH/CeO_2_-NPs

Cytotoxicity of GH/CeO_2_-NPs was evaluated against normal RAW 264.7 macrophage cells obtained from Sigma, St. Louis, MO, USA, using MTT assay. Cells (5 × 10^4^ cell/well) were seeded in Dulbecco’s modified eagle’s medium (DMEM; Gibco) containing 10% fetal bovine serum, 100 μg/mL streptomycin, and 100 U/mL penicillin in a 96-well plate for 24 h incubation at 37 °C in 5% CO_2_ to reach a cell density of 80% confluence and treated with nanoparticles of 0, 18.7, 37.5, 75, 150, 300, and 400 µg/mL. The proposed concentrations were chosen based on previous research performed with CeO_2_ NPs [44]. Untreated cells and 5% DMSO-treated cells were assayed as negative and solvent control, respectively. After 24 h incubation, 20 µL MTT dye (5 mg/mL) was pipetted in all wells, including wells with untreated cells, and incubated for 4 h. The medium was aspirated, and 100 µL of DMSO was added to all the wells to dissolve the formazan crystals and measured after an hour at 570 nm using a microplate reader. The percentage of cell viability was calculated from the ratio of the absorbance of treated cells with respect to those untreated.

#### 4.3.8. Statistical Analysis

All values were reported as a mean ± SD. Statistical analysis of data was carried out by analysis of Student’s *t*-test comparing 2 groups for continuous variables with normal distributions, and one-way analysis of variance (ANOVA) was used for multiple groups, with Tukey’s post hoc analysis. p values less than 0.05 were considered statistically significant.

## 5. Conclusions

The cerium nanoparticles were synthesized effectively through a sol-gel synthesis route using ammonium hydroxide as a reducer. Synthesized nanoparticles present stability, nano-size, amorphous nature, spherical shape, and GH encapsulation into cerium, indicating an excellent delivery ability. Aggregation of proteins, tau oligomers and amyloid-beta oligomers, acetylcholine, and butyrylcholinesterase are considered the main cause of mitochondrial dysfunction, producing neuronal toxicity and resulting in Alzheimer’s disease. The induction of the β-amyloid structure in the BSA-ribose system was suppressed in the presence of CeO_2_-NPs, GH, and GH/CeO_2_-NPs, and donepezil. GH/CeO_2_-NPs were observed to be more potent in impeding the development of aggregates in the BSA-ribose system than other samples. The combined GH and Ce nanoparticles, possibly due to a synergistic effect, efficiently improve aggregation proteins to prevent or delay the formation of fibrillar species or aggregated amyloid proteins, which are characteristics of amyloid fibril formation in AD disease. Tau oligomers and amyloid-β oligomers, by improving GH biocompatibility and biodegradability, consequently enhance the therapeutic efficacy. The neuroprotective effects of GH/CeO_2_-NPs were confirmed in AlCl_3_-induced AD to increase their capacity to improve cognitive functions and memory abilities. In addition, improved neurotransmitters modify by inhibiting AChE and BuChE activities, indicating the neuromodulator effect of nanoparticles. The results provide relevant insights into a new potential anti-neurodegenerative that can decrease the severity of Aβ1–42 fibril formation. These findings showed that nanoparticles have played a promising role in drug anti-aggregating against neurodegenerative disorders.

## Figures and Tables

**Figure 1 ijms-25-04262-f001:**
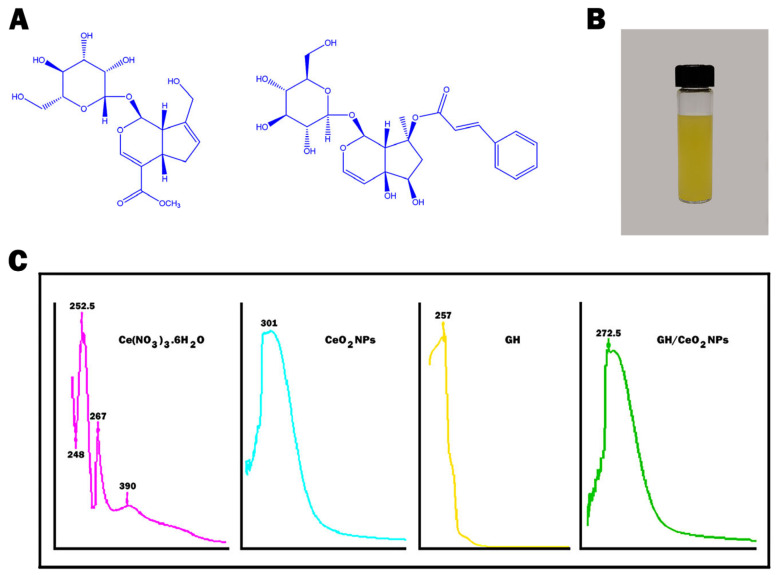
The chemical structure of (**A**) geniposide and harpagoside (lilac); (**B**) Formation of CeO_2_ NPs (yellow); (**C**) UV–visible spectra of Ce(NO_3_)3.6H_2_O, CeO_2_ NPs GH and GH/CeO_2_ NPs (green).

**Figure 3 ijms-25-04262-f003:**
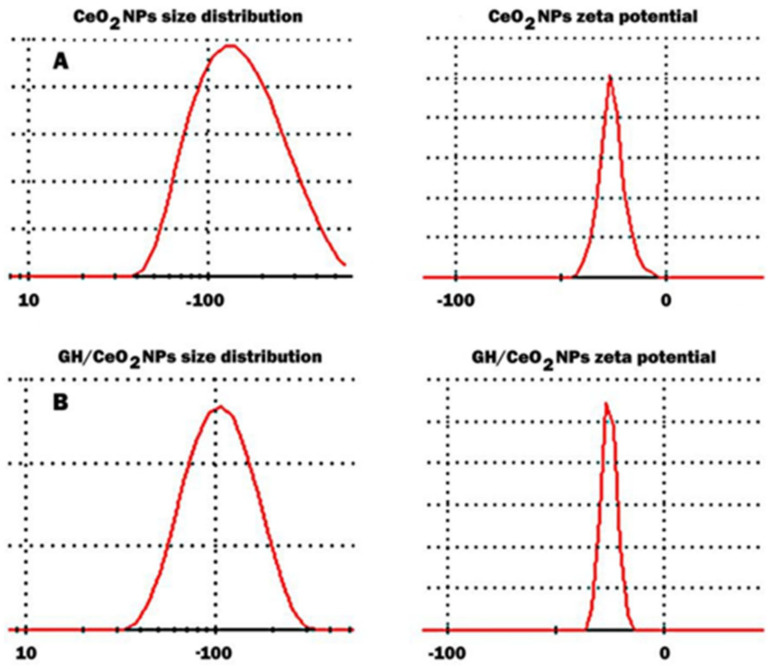
Size distribution and zeta potential of CeO_2_ NPs (**A**); size distribution and zeta potential of GH/CeO_2_ NPs (**B**).

**Figure 4 ijms-25-04262-f004:**
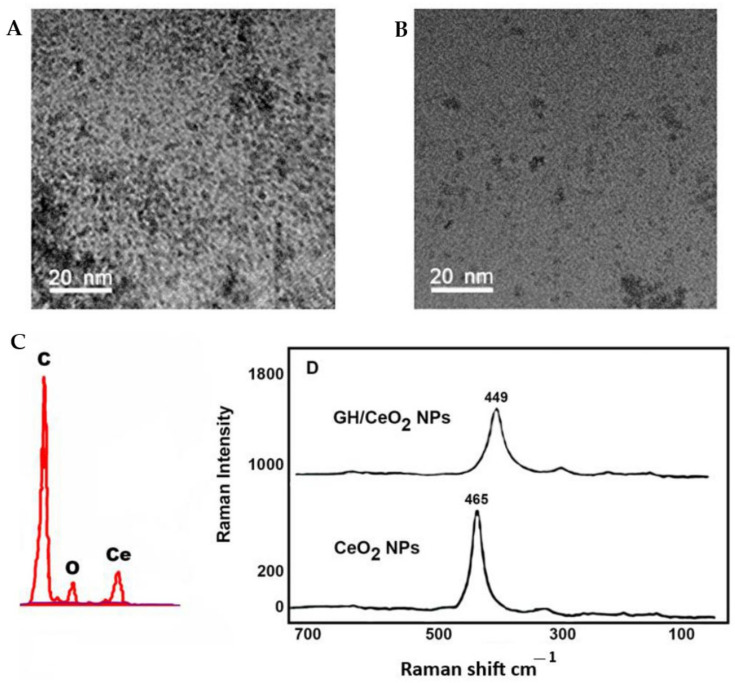
Images of synthesized nanoparticles: (**A**) CeO_2_ NPS (20 nm); (**B**) GH/CeO_2_ NPs (20 nm); (**C**) GH/CeO_2_ NPs (50 nm); X-ray diffraction analysis of GH/CeO_2_ NPs. The images were taken at a resolution of 15,000× and a scale bar of 20 and 50 nm. (**D**) Raman spectrum of GH/CeO_2_ NPs.

**Figure 5 ijms-25-04262-f005:**
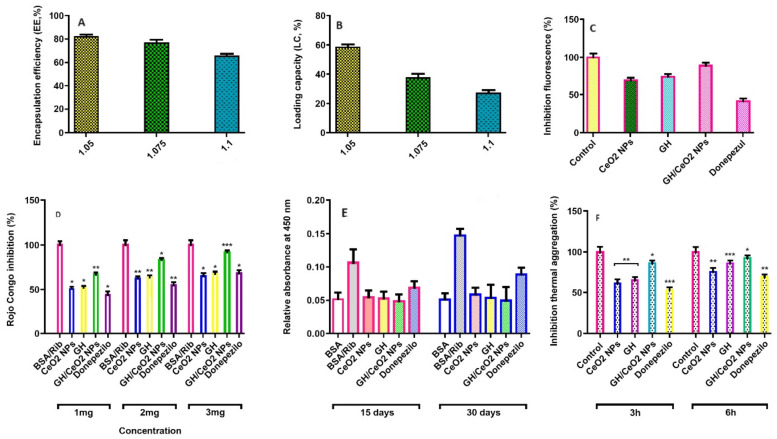
Determination of encapsulation efficiency (**A**); loading capacity (**B**) obtained at a ratio of 1:0.5, 1.075, and 1:1 of cerium/GH; fibrillation results in ThT test assay at Ex-440 nm and Em-490 nm generated by CeO_2_ NPs, GH, GH/CeO_2_ NPs, and donepezil in ribose incubated BSA (**C**); the effect of nanoparticles in vitro formation of amyloid-β products—Congo red dye in BSA-ribose glycation at 490 nm (**D**); Aβ aggregation measurement at 15 days and 30 days BSA/Ribose incubation by turbidimetric analysis at 450 nm (**E**); Aβ aggregation measurement at 3 and 6 h in BSA/glucose incubation by thermal assay (**F**); data are expressed as the mean ± SD; asterisks (*) indicate significant statistical differences between treated and untreated control * *p* < 0.05; ** *p* < 0.01; *** *p* < 0.001.

**Table 1 ijms-25-04262-t001:** AChE and BuChE inhibitory activities from CeO_2_NPs, GH and GH/CeO_2_NPs.

Sample	Inhibition (%)	
	ACHE	BuChE
CeO_2_NPs		
10 µg	33.9	37.4
20 µg	36.9	39.2
30 µg	41.7	44.5
GH		
10 µg	43.6	43.4
20 µg	47.4	46.1
30 µg	49.7	49.7
GH/CeO_2_NPs		
10 µg	49.4	57.2
20 µg	52.3	61.3
30 µg	67.0	63.1
Donepezil		
10 µg	58.8	55.4
20 µg	65.8	67.2
30 µg	88.7	84.0

GH/CeO_2_-NPs showed dose-dependent inhibition on AChE and BuChE.

**Table 2 ijms-25-04262-t002:** Effect of BSA-ribose incubated with GH, CeO_2_ NPs, GH/CeO_2_ NPs, and donepezil on zeta potential, average diameter, polydispersity index, and fluorescence for 30 days.

Sample	Zeta Potential (mV)	Average Diameter (nm)	Polydispersity Index	Fluorescence (A.U)
BSA	−8.41	9.2	0.13	8.6
BSA/ribose	−34.6	4355	1.0	74.9
BSA/ribose + GH	−22.8	2078	0.397	42.6
BSA/ribose + CeO_2_ NPs	−27.2	2482	0.412	50.2
BSA/ribose + GH/CeO_2_ NPs	−19.1	1765	0.365	35.1
BSA/ribose + Donepezil	−29.3	3223	0.638	61.5

**Table 3 ijms-25-04262-t003:** Aggregation of amyloid β1–42 in the presence of CeO_2_ NPs, GH, and GH/CeO_2_ NPs. Results represent the fluorescence intensity and inhibition (%) as a function of time with 4, 6, 8, and 12 h.

Th Assay β1–42 Aggregation Fluorescence in RFU/Inhibition (%)					
Sample	4 h	6 h	8 h	12 h	24 h
Amyloid β1–42	6200	7800	9865	11,750	12,120
CeO_2_ NPs	5743 (7.4)	6821 (12.5)	6832 (30.7)	5131 (56.3)	2253 (81)
GH	5282 (14.8)	6120 (21.5)	6432 (34.7)	4765 (59)	1876 (84)
GH/CeO_2_ NPs	5181 (16.4)	5623 (27.9)	6138 (37.7)	4322 (63.2)	1087 (91)

**Table 4 ijms-25-04262-t004:** The results of the aggregation of tau protein in the presence of CeO_2_ NPs, GH, and GH/CeO_2_ NPs represent the fluorescence intensity and inhibition (%).

Th Assay Tau Aggregation Fluorescence in RFU/Inhibition (%)						
Sample	20 h	40 h	60 h	80 h	100 h	120 h
Tau protein	2500	4100	6341	8641	9851	11,500
CeO_2_ NPs	2000 (20)	2300 (43.9)	2650 (58.2)	3284 (62)	3648 (63)	3401 (70.4)
GH	1832 (26.7)	2125 (48.1)	2360 (62.8)	2460 (71)	2619 (73.4)	2671 (77)
GH/CeO_2_ NPs	1654 (33.8)	1720 (58)	1964 (69)	2000 (76.8)	2003 (79.5)	2004 (83)

Results are as a function of time with 40, 60, 80, 100, and 120 h.

## Data Availability

Data are contained within the article.

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
