# Peer review of "Geniposide and Harpagoside Functionalized Cerium Oxide Nanoparticles as a Potential Neuroprotective"

_ijms, 2024, doi:10.3390/ijms25084262_

Round 1

Reviewer 1 Report (New Reviewer)

Comments and Suggestions for Authors

The submitted article "Geniposide and Harpagoside Functionalized Cerium Oxide Nanoparticles as a Potential Neuroprotective" by Gomez et al. contains numerous experimental failures. Additionally, it reveals that the authors lack familiarity with enzymatic assays and the process of encapsulation, demonstrating a general misunderstanding of the above mentioned topic. Therefore, the manuscript should not be accepted for publication.

Regarding minor remarks:

"The most common type of dementia is Alzheimer's disease (AD), which is related to aggregating proteins, oxidative stress, and acetylcholinesterase."

This sentence needs further explanation and supporting references. You could expand on how Alzheimer's disease is associated with protein aggregation, oxidative stress, and the role of acetylcholinesterase in the pathology of the disease. Providing specific references to reputable sources would strengthen the statement.

Please check this statement: "Amyloid β (Aβ) inhibition of AChE, and butylcholines are three important parameters that contribute to the initiation and progression of Alzheimer's disease."

What are butylcholines? If you mean 'butyrylcholine' instead of 'butylcholines,' you should know that it is a synthetic neurotransmitter."

This statement requires verification and supporting references. You might want to clarify the roles of amyloid β (Aβ), acetylcholinesterase (AChE) inhibition, and butylcholines in the context of Alzheimer's disease. Providing references to scientific literature will enhance the credibility of the statement.

"And a total encapsulation of GH in cerium."??? This statement appears unclear!!!

Comments on the Quality of English Language

The English in the manuscript requires significant changes!

Author Response

Reviewer1

The submitted article "Geniposide and Harpagoside Functionalized Cerium Oxide Nanoparticles as a Potential Neuroprotective" by Gomez et al. contains numerous experimental failures. Additionally, it reveals that the authors lack familiarity with enzymatic assays and the process of encapsulation, demonstrating a general misunderstanding of the above mentioned topic. Therefore, the manuscript should not be accepted for publication.

Regarding minor remarks:

"The most common type of dementia is Alzheimer's disease (AD), which is related to aggregating proteins, oxidative stress, and acetylcholinesterase."

This sentence needs further explanation and supporting references. You could expand on how Alzheimer's disease is associated with protein aggregation, oxidative stress, and the role of acetylcholinesterase in the pathology of the disease. Providing specific references to reputable sources would strengthen the statement.

The paragraph was changed by Alzheimer's disease is associated with protein aggregation, oxidative stress, and the role of acetylcholinesterase in the pathology of the disease.

 But It is not possible to add a reference because it is in the abstract

Please check this statement: "Amyloid β (Aβ) inhibition of AChE, and butylcholines are three important parameters that contribute to the initiation and progression of Alzheimer's disease."

The statement was changed

What are butylcholines? If you mean 'butyrylcholine' instead of 'butylcholines,' you should know that it is a synthetic neurotransmitter."

The paragraph was changed

This statement requires verification and supporting references. You might want to clarify the roles of amyloid β (Aβ), acetylcholinesterase (AChE) inhibition, and butylcholines in the context of Alzheimer's disease. Providing references to scientific literature will enhance the credibility of the statement.

The roles of amyloid β (Aβ), acetylcholinesterase (AChE) inhibition, and butylcholines in the context of Alzheimer's disease were clarified

"And a total encapsulation of GH in cerium."??? This statement appears unclear!!!

Encapsulation of GH in cerium was removed

Reviewer 2 Report (New Reviewer)

Comments and Suggestions for Authors

The current manuscript is an interesting experimental study on the development of geniposide and harpagoside loaded within cerium oxide nanoparticles for neuroprotection purposes. It appears to be overall well-done, with many relevant assays having been performed. Hence, I only advise on the following alterations before acceptance for publication:

- In the introduction section, more should be said about current therapies, with the mention of specific drug molecules that are part of common treatment regimens, and their limitations;

- The limitations of using cerium as part of nanoparticle composition should be mentioned: what are the possible toxic effects? Here are some references that mention possible serious complications: https://pubmed.ncbi.nlm.nih.gov/17132603/; https://www.nature.com/articles/srep15613; https://pubmed.ncbi.nlm.nih.gov/24987704/; in light of this information, the authors should argument of the risk-benefit ration of using cerium NP;

- Figure quality (resolution) should be improved, namely figures 1, 2, 3, 4, 5, and 6;

- Tables are inserted as images, and this should be corrected (they should be inserted as actual tables), namely tables 1, 2, 3, and 4;

- Equations 39 should be formatted in a way they are understandable, they are currently too big and confusing; same for the equations on lines 636 (brackets and spaces missing), and 670;

- English language should be reviewed throughout the whole manuscript, as should errors (typos) such as the missing bracket on line 20, or the missing space and unnecessary full stop mark on the title on line 204;

- An abbreviation list is missing and should be added.

Comments on the Quality of English Language

Moderate editing of English language required.

Author Response

Reviewer 2

The current manuscript is an interesting experimental study on the development of geniposide and harpagoside loaded within cerium oxide nanoparticles for neuroprotection purposes. It appears to be overall well-done, with many relevant assays having been performed. Hence, I only advise on the following alterations before acceptance for publication:

- In the introduction section, more should be said about current therapies, with the mention of specific drug molecules that are part of common treatment regimens, and their limitations;

In the introduction is broad current therapies, with with the mention of the common treatment regimens, and their limitations.

- The limitations of using cerium as part of nanoparticle composition should be mentioned: what are the possible toxic effects? Here are some references that mention possible serious complications: https://pubmed.ncbi.nlm.nih.gov/17132603/; https://www.nature.com/articles/srep15613; https://pubmed.ncbi.nlm.nih.gov/24987704/; in light of this information, the authors should argument of the risk-benefit ration of using cerium NP;

Se describen los efectos toxicos y lpos beneficios del cerio y de  las nanopsaarticulas de cerio

The toxic effects and benefits of cerium and cerium nanopsaarticulates are described.

- Figure quality (resolution) should be improved, namely figures 1, 2, 3, 4, 5, and 6;

Figures 1-6 have the resolution increased

- Tables are inserted as images, and this should be corrected (they should be inserted as actual tables), namely tables 1, 2, 3, and 4;

Tables 1-4  were isnserted as actual tables  inside the manuscript. But on the platform   cannot upload the tables alone so they must be in ZIP along with the figures

- Equations 39 should be formatted in a way they are understandable, they are currently too big and confusing; same for the equations on lines 636 (brackets and spaces missing), and 670;

The equations were formated

- English language should be reviewed throughout the whole manuscript, as should errors (typos) such as the missing bracket on line 20, or the missing space and unnecessary full stop mark on the title on line 204;       

English language was revised

- An abbreviation list is missing and should be added.

The Journal format does not require abbreviations

Reviewer 3 Report (Previous Reviewer 2)

Comments and Suggestions for Authors

Please answer the comments of the reviewer one by one, point-by-point. Otherwise, it will be difficult to proceed with the review process.

Comments on the Quality of English Language

Please answer the comments of the reviewer one by one, point-by-point. Otherwise, it will be difficult to proceed with the review process.

Author Response

Reviewer 3

There are numerous typos in the main text, so the reviewer cannot review further. The authors should confirm those errors/mistakes and resubmit again. Article has serious flaws, additional experiments needed, research not conducted correctly.

-On page 11, "In ours sample" should be changed to 'In our sample".

In ours sample was changed by  In our sample

-We got values of PDI of less than 0.1, indicating that the nanoparticles were near mono-disperse. Z-average sizes. This sentence has a serious mistake.

The topic was explained more fully.

-Amyloid-b should be changed to Amyloid-beta.

Amyloid-b was changed by Amyloid-beta.

-These results are agreed with thioflavin T observations (Fig 5C). This sentence is not correct in grammatical English.

The sentence was changed by These results agree with the data obtained in the  thioflavin T-test

-In Abstract, "selenium nanoparticles (CeO2 NPs)" is correct?

selenium nanoparticles was removed

Any other mistakes are found in the main text.

The mistakes were corrected

Round 2

Reviewer 1 Report (New Reviewer)

Comments and Suggestions for Authors

These are not an answers to the questions in the last reviewer's report. In any case this manuscript should not be accepted for publication.

Author Response

The submitted article "Geniposide and Harpagoside Functionalized Cerium Oxide Nanoparticles as a Potential Neuroprotective" by Gomez et al. contains numerous experimental failures. Additionally, it reveals that the authors lack familiarity with enzymatic assays and the process of encapsulation, demonstrating a general misunderstanding of the above mentioned topic. Therefore, the manuscript should not be accepted for publication.

Regarding minor remarks:

"The most common type of dementia is Alzheimer's disease (AD), which is related to aggregating proteins, oxidative stress, and acetylcholinesterase."

This sentence needs further explanation and supporting references. You could expand on how Alzheimer's disease is associated with protein aggregation, oxidative stress, and the role of acetylcholinesterase in the pathology of the disease. Providing specific references to reputable sources would strengthen the statement.

The paragraph was changed by Alzheimer's disease is associated with protein aggregation, oxidative stress, and the role of acetylcholinesterase in the pathology of the disease.

 But It is not possible to add a reference because it is in the abstract

Please check this statement: "Amyloid β (Aβ) inhibition of AChE, and butylcholines are three important parameters that contribute to the initiation and progression of Alzheimer's disease."

The statement was changed

What are butylcholines? If you mean 'butyrylcholine' instead of 'butylcholines,' you should know that it is a synthetic neurotransmitter."

The paragraph was changed

This statement requires verification and supporting references. You might want to clarify the roles of amyloid β (Aβ), acetylcholinesterase (AChE) inhibition, and butylcholines in the context of Alzheimer's disease. Providing references to scientific literature will enhance the credibility of the statement.

The roles of amyloid β (Aβ), acetylcholinesterase (AChE) inhibition, and butylcholines in the context of Alzheimer's disease were clarified

"And a total encapsulation of GH in cerium."??? This statement appears unclear!!!

Encapsulation of GH in cerium was removed

Reviewer 3 Report (Previous Reviewer 2)

Comments and Suggestions for Authors

It is not reasonable to reconsider a revised manuscript that has been rejected in a previous review. If I would like be an editor-in-chief, I would categorize it as 'rejected with encouragement for resubmission.' Additionally, authors must provide responses to reviewer comments on a point-by-point basis for further review.

Comments on the Quality of English Language

It is not reasonable to reconsider a revised manuscript that has been rejected in a previous review. If I would like be an editor-in-chief, I would categorize it as 'rejected with encouragement for resubmission.' Additionally, authors must provide responses to reviewer comments on a point-by-point basis for further review.

Author Response

There are numerous typos in the main text, so the reviewer cannot review further. The authors should confirm those errors/mistakes and resubmit again. Article has serious flaws, additional experiments needed, research not conducted correctly.

Typographical errors were corrected

-On page 11, "In ours sample" should be changed to 'In our sample".

The phrase was changed

-We got values of PDI of less than 0.1, indicating that the nanoparticles were near mono-disperse. Z-average sizes. This sentence has a serious mistake.

The phrase was changed to values of  0.1 or  lower is ideal instead values  above  0.1 indicates more  species are present in solution [20].

-Amyloid-b should be changed to Amyloid-beta.

Amyloid-b was changed to changed to Amyloid-beta.

-These results are agreed with thioflavin T observations (Fig 5C). This sentence is not correct in grammatical English.

The phrase was changed

-In Abstract, "selenium nanoparticles (CeO2 NPs)" is correct?

Se was modified by CeO2

-Any other mistakes are found in the main text.

Mistakes were revised

Round 3

Reviewer 1 Report (New Reviewer)

Comments and Suggestions for Authors

The introductory section in the submitted paper clearly reflects a lack of understanding of the field the authors were engaged in. There are numerous errors throughout the paper, including those related to the experimental section as well as the abstract and introduction. The paper fails to meet even the minimum criteria for publication in this journal.

TEM images of amyloid aggregates are missing!

Additionally, '' inhibition of acetylcholinesterase and butyryl thiocholine"  line 665 - did the authors understand the differences between these two things? The first is an enzyme and can be tested, and the second is a substrate and cannot be tested. Due to many similar errors, this reviewer believes that this paper must be rejected.

Reviewer 3 Report (Previous Reviewer 2)

Comments and Suggestions for Authors

Thank you for your revised manuscript. The referee reviewed your manuscript and raised some comments as below:

1. The title of your paper is grammatically incorrect. It should be either "~~a potential neuroprotective agent" or "~~~ potential neuroprotective agents". 

2. Even in the ABSTRACT section, specific figures were not presented compared to positive or negative controls. 

3. The presentation of all other data and/or information below is insufficient. 

This manuscript is a resubmission of an earlier submission. The following is a list of the peer review reports and author responses from that submission.

Round 1

Reviewer 1 Report

Comments and Suggestions for Authors

Row 66-67 "Butyrylcholinesterase (BuChE) is a serine hydrolase that catalyzes the hydrolysis of non-choline esters, the neurotransmitter acetylcholine and choline". Butyrylcholinesterase catalyzes he hydrolysis of choline?

Alzheimer's disease was induced by using AlCl3? Which method were used to establish the disease induced?

Please display the information about the toxicity of these nanoparticles.

I asking the researcher about the tests that have been used to certify that the geniposide has been encapsulated or has been fixed on NPs surface.

Why they choose to  give AlCl3 as Alzheimer inductor. AlCl3 induces neurotoxicity.

Author Response

Please

Comments and Suggestions for Authors 

Review 1

Row 66-67 "Butyrylcholinesterase (BuChE) is a serine hydrolase that catalyzes the hydrolysis of non-choline esters, the neurotransmitter acetylcholine and choline". Butyrylcholinesterase catalyzes he hydrolysis of choline?

Butyrylcholinesterase (BChE) is a nonspecific cholinesterase enzyme that hydrolyzes choline-based esters. BChE plays a critical role in maintaining normal cholinergic function like acetylcholinesterase.

We had a mistake, the paragraph was edited.

Alzheimer's disease was induced by using AlCl3? Which methods were used to establish the disease induced?

We determine the induction of AD in mice using the following   neurobehavioral models such as:
T-maze Test, Novel objection Recognition (NOR) Test and Exploratory Behavior in Open-field Test.
Also,  the model of  Detection of Amyloid Aggregates in  plasma was used.

With all these experiments we confirmed the induction of AD in the animals

Please display the information about the toxicity of these nanoparticles

At the concentrations used in this work, no toxic effects were observed (data not presented in the article)

I asking the researcher about the tests that have been used to certify that the geniposide has been encapsulated or has been fixed on NPs surface.

We did an  initial test with selenium nanoparticles functionalized with Geniposide and another with Harpagoside. The results indicated that Se NPs were loaded with each iridoid because both contain OH groups in their structure which bind to the cerium metal nanoparticle (data not presented in the article because it was not the objective of the work).

Why they choose to  give AlCl3 as Alzheimer inductor. AlCl3 induces neurotoxicity.

Chronic administration of AlCl3 at various levels in mice has been utilized in various investigations to mimic the physiology of Alzheimer’s disease. In our study, AlCl3 was selected based on previous reports because of the high rate of induction and low mortality .

see the attachment

Reviewer 2 Report

Comments and Suggestions for Authors

Thank you for submitting your manuscript to IJMS. The reviewer carefully read your manuscript and raised some concerns, as below:

1. In Abstract, lines 22, 24, and 27, you mentioned "selenium nanoparticles and GH/ Se NPs." Did you mean a positive control or a negative control? 

2. In the main text, there are no positive and/or negative control(s). Please add them. 

3. In Fig. 6, please add image data. 

4. The title is too long. Please simplify.

5. The current nanoparticles are not toxic to any types of cells or tissues?

Comments on the Quality of English Language

Extensive editing of English language required

Author Response

Please

Thank you for submitting your manuscript to IJMS. The reviewer carefully read your manuscript and raised some concerns, as below:

1.In Abstract, lines 22, 24, and 27, you mentioned "selenium nanoparticles and GH/ Se NPs." Did you mean a positive control or a negative control? 

The donepezil used as positive control was added in the abstract.

2.In the main text, there are no positive and/or negative control(s). Please add them. 

The donepezil used as positive control was added in the text.

  1. In Fig. 6, please add image data. 

In Fig. 6, were add image data.

Fig6. Effect of GH, CeO2 NPs and GH/CeO2 NPs to determine behavioral status in experimental AD mice.   (A)    T-maze tests show spatial perceptive ability  as percentage   of number of arm visits for 5 min ;(B) total squares crossed (5 min) in open filed .   In the novel object recognition test  the space perceptive ability is determined as the  percentage of the identification  accounts a  rodent  in each object for   5 min  (C). In the  object recognition test   abilities for familiar  and new object  are significantly different as determined by Student’s t-test (*p < .05). (D) ELISA test reducing  Aβ deposition after I GH/CeO2 NPs treatment in plasma  of  AlCl3 mice. Effect of samples  on plasma amyloids AB1-42  (D), Data are expressed as the mean ± SD   ***p < 0.001, **p < 0.01, and *p < 0.05 compared with AlCl3 group.

  1. The title is too long. Please simplify.
    The title was reduced.

Geniposide and Harpagoside    Functionalized  Cerium Oxide Nanoparticles   Exert Neuroprotective Effects Against Protein Aggregation,   Acetylcholinesterase, Butyrylcholinesterase, Amyloid-beta,   and  Cognitive Impairment  

  1. The current nanoparticles are not toxic to any types of cells or tissues?

At the concentrations used in this work, no toxic effects were observed.

Extensive editing of English language required.

English language was revised.

The references were diminished

see the attachment

Round 2

Reviewer 2 Report

Comments and Suggestions for Authors

There are numerous typos in the main text, so the reviewer cannot review further. The authors should confirm those errors/mistakes and resubmit again. Article has serious flaws, additional experiments needed, research not conducted correctly.

-On page 11, "In ours sample" should be changed to 'In our sample".

-We got values of PDI of less than 0.1, indicating that the nanoparticles were near mono-disperse. Z-average sizes. This sentence has a serious mistake.

-Amyloid-b should be changed to Amyloid-beta.

-These results are agreed with thioflavin T observations (Fig 5C). This sentence is not correct in grammatical English.

-In Abstract, "selenium nanoparticles (CeO2 NPs)" is correct?

-Any other mistakes are found in the main text.

[Final decision] Reject

Comments on the Quality of English Language

There are numerous typos in the main text, so the reviewer cannot review further. The authors should confirm those errors/mistakes and resubmit again. Article has serious flaws, additional experiments needed, research not conducted correctly.

Author Response

(The authors gave the same response as above.)
